# Aging Resistance of Biocomposites Crosslinked with Silica and Quercetin

**DOI:** 10.3390/ijms221910894

**Published:** 2021-10-08

**Authors:** Anna Masek, Olga Olejnik

**Affiliations:** Faculty of Chemistry, Institute of Polymer and Dye Technology, Lodz University of Technology, Ul. Stefanowskiego 12/16, 90-924 Lodz, Poland; olga.olejnik@dokt.p.lodz.pl

**Keywords:** quercetin, stabilizer, ENR, biocomposite

## Abstract

This research focuses on revealing the double role of quercetin accompanied by silica in epoxidized natural rubber. A crosslinking ability with antioxidative properties exists and reveals the dependence of these functions on quercetin content. Here, the aging resistance of self-healable biocomposites was analyzed. The self-healing properties were presented in our previous work. The stabilizing effect of quercetin applied as a crosslinking agent has been studied in epoxidized natural rubber with a 50 mol% of epoxidation (ENR-50). Some of five -OH moiety groups existing in the quercetin structure are able to react with epoxy rings of ENR-50 and cure this elastomer, whereas other free hydroxyl groups can donate the hydrogen molecule to a radical molecule, stabilizing it. The aging resistance of prepared composites was estimated by mechanical tests conducted before and after different types of aging, as well as by differences in color and surface energy between aged and un-aged samples. Changes within the oxygen function, which occurred as a result of the aging process, were observed using FT-IR absorbance spectroscopy. Furthermore, the impact of quercetin content on composites’ thermal stability was investigated by thermogravimetry (TGA). According to the results, a proper dose of quercetin can act as a crosslinker and antioxidant in ENR-50 at the same time.

## 1. Introduction

Flora offers an enormous amount of beneficial phytochemicals which can be applied in polymer technology [1,2]. In recent years, a large group of natural antioxidants [3], especially flavonoids, have been examined in terms of utilization as natural additives dedicated to different polymers, including thermoplastics [4,5] and elastomers [6,7]. One of the most popular and well examined flavonoids is quercetin, naturally and widely occurring in many types of fruits, vegetables, spices and herbs [8]. Unlike many synthetic stabilizers, this substance is safe for human beings because it is characterized not only by antioxidant properties but also therapeutic properties, namely, its anticarcinogenic, anti-inflammatory and anti-viral effects [9]. As a result of its origin and its biodegradability, this flavonoid can be used in biomaterials as an environmentally friendly additive, which is able to maintain the pro-ecological character of biocomposites [10].

Quercetin is a type of flavonol (a subclass of flavonoids [11]) characterized by a 3-hydroxyflavone backbone. The structure of this flavonoid contains five hydroxyl groups (Figure 1), which play a key role in the oxidation process, preventing other materials, including polymeric matrix, from oxidative degradation [9]. This substance is able to chelate metal ions as well as act as a free radical scavenger [12]. The antioxidative effect of quercetin was investigated in different polymeric matrixes, including poly (vinyl alcohol) [13], ethylene–norbornene copolymer (Topas^®^) [14] and more complex composites based on biopolymers, such as polylactide (PLA) and polycaprolactone (PCL) [15].

This popular flavonoid is also able to absorb a wide range of UV light, preventing polymers from photodegradation [16]. The photo-stabilizing effect of quercetin was investigated, inter alia, by Arrigo et al. in biodegradable polymers, including PLA [4], and by Samper et al. in polypropylene [5]. The latter researchers also confirmed that this popular flavonoid can be applied as a thermal stabilizer.

The coloring effect of quercetin was investigated in an ethylene–norbornene copolymer [14]. In the same research, it was confirmed that this flavonoid can be used as an aging indicator, because its color changes during the aging process. A similar effect was also recognized in ENR/PLA blends [10].

The crosslinking properties of quercetin were also investigated. It has been confirmed that this flavonoid exhibits a curing effect in polymer blends consisting of epoxidized natural rubber (ENR) and polycaprolacone (PCL) [17]. In our previous research, the crosslinking ability of quercetin was also confirmed in the ENR-50 matrix in the presence of silica [18]. This article is a continuation of the previous publication, which focused on the self-healing ability of ENR-50-composites. Here, we are focused on applying a combination of quercetin and silica, not only as a crosslinking agent, but also a stabilizer dedicated to the ENR-50 elastomer to obtain pro-ecological and safe biomaterials with an improved aging resistance.

## 2. Results and Discussion

### 2.1. Thermogravimetry Analysis

As mentioned above, the stabilization effect can occur in many instances, not only in preventing material from the oxidation process, but also, for example, in inhibiting its thermal degradation. Replacing the conventional crosslinking agent, dicumyl peroxide (DCP), with only silica or a combination of quercetin and silica, leads to the improvement in the material’s thermal stability in comparison with ENR/DCP, which is depicted in Figure 2 and Table 1. The ENR-50 composite cured by only 15 phr of silica caused a 2% loss of its mass at a higher temperature than ENR/DCP at about 18 °C and around 7 °C compared to uncured ENR-50. A similar positive effect of silica addition was also observed by Xu et al. [19], where the addition of 20 phr of silica to pure ENR-40 caused an improvement in the material’s thermal stability and changed the temperature by a 5% material loss in mass from 218.9 °C to 313.3 °C. According to the same publication, epoxidized natural rubber with a higher amount of active filler is more vulnerable to thermal degradation than uncured ENR. This may be a result of greater stiffness resulting from a higher number of crosslinks, which is a weak point during elastomer degradation; it means that the proper amount of silica can act not only as a curing agent, but also as a thermal stabilizer.

Quercetin was investigated as a thermal stabilizer dedicated to polypropylene (PP) by Samper et al. [5]. According to the paper, the 0.25% of quercetin in the polypropylene matrix resulted in an improvement in the material’s thermal stability and moved the decomposition onset temperature of PP from 265.9 °C to 301.1 °C. In our research, adding 2 phr of quercetin to the ENR-50 matrix as a crosslinking agent did not change the thermal stability compared to the results of pure, uncured ENR. Nevertheless, ENR-50 composites crosslinked with 2 phr of quercetin were characterized by a better thermal stability than conventionally crosslinked ENR (ENR/DCP). ENR/Q2 material starts to lose 2% of its mass at a higher temperature, of about 10 °C, than ENR/DCP composites. The addition of a higher amount of quercetin does not have a positive influence on the thermal stability of ENR-50; this means that such a composite starts to decompose at 328 °C, which is a lower temperature than for pure uncured ENR-50 or conventionally crosslinked ENR/DCP. The crosslinks created in the ENR-50 matrix by quercetin are weak points and not as strong as bonds formed by dicumyl peroxide (DCP). Therefore, the whole material is more vulnerable to thermal degradation, and a higher amount of quercetin cannot play the role of thermal stabilizer and crosslinker at the same time.

ENR-based composites cured with a combination of 15 phr of silica and 2 phr of quercetin revealed a better thermal stability than the ENR-based materials crosslinked with dicumyl peroxide, which is a conventional curative. A higher amount of quercetin (4 phr) in ENR/SIL/Q4, similarly to ENR/Q4, contributes to forming a higher number of weak bonds, which are more vulnerable to thermal degradation. Replacing DCP by 2 phr of quercetin, 15 phr of silica, or by a mixture of presented components, resulted in a better thermal stability of the composites in comparison to conventionally cured ENR. In the presented composites, silica is mostly responsible for better thermal stability.

### 2.2. Static Mechanical Tests

The static mechanical tests, which were conducted before and after different types of aging processes, presented the stability of the created composites. According to Figure 3, ENR-50-based materials cured with 2 phr of DCP, or with the same mass ratio of quercetin, are characterized by the lowest aging stability and the aging coefficient after thermo-oxidative aging as well as solar aging amounting to approximately 0.5. On the other hand, the composites containing a higher amount of quercetin were characterized by significantly higher tensile strength (TS) after different aging processes. This means that the crosslinking process occurred in ENR-50 under selected conditions, including raised temperature (T = 70 °C) with no radiation (i.e., thermo-oxidative aging) and low temperature (T = −10 °C) with solar radiation (i.e., solar aging). The best aging stability was detected in the case of ENR/SIL composites and ENR/SIL/Q2, and for these materials, the aging coefficient fluctuated around 1; however, slight differences between them were noticeable. In particular, ENR/SIL composites revealed a delicate deterioration in TS and Eb after thermo-oxidative aging, whereas ENR/SIL/Q2 material was characterized by an improved TS. On the other hand, solar radiation only caused a slight improvement in both types of these composites, which also indicates that the slight crosslinking process could occur under aging conditions. The uncured ENR-50 after thermo-oxidative aging could not be tested because of a high level of stickiness as a result of degradation; therefore, a sample of pure ENR-50 after thermo-oxidative aging is missing. However, low temperature and solar aging did not significantly affect the mechanical properties of this material.

In spite of the fact that ENR/SIL and ENR/SIL/Q2 composites revealed similar aging stability, adding 2 phr of quercetin to ENR-50 with 15 phr of silica, resulted in a higher tensile strength of about 1.1 ± 0.2 MPa, but also lower elongation at break of about 290 ± 50% due to the rigid aromatic ring present in the structure of quercetin and higher amount of crosslinks, which is also visible in Appendix A. A higher dose of quercetin (4 phr) caused the highest rigidity because of an increased number of crosslinks, which grew during the aging processes.

Therefore, ENR-50-based composite cured with a combination of only 2 phr of quercetin and 15 phr of silica has similarly stable-aging status in comparison to ENR/SIL and is more resistant to aging than ENR/SIL/Q4. It means that ENR-50, cured with a combination of 2 phr of quercetin and 15 phr of silica, provides the best compromise between tensile strength (TS) and aging stability.

### 2.3. Surface Free Energy (SFE) Measuements

The contact angle and surface free energy measurements of tested ENR-50-based elastomers are presented in Figure 4. According to these results, the surfaces of all tested materials were quite hydrophobic, and the contact angles measured using water were above 100° in all cases. The hydrophobicity decreased after the aging processes, but water was still unable to dampen the surface of the samples. Additional liquids, including diiodomethane and ethylene glycol, were used to calculate the surface free energy (SFE). Based on the SFE values, it can be noted that uncured ENR-50 was characterized by an SFE of 23.57 ± 2.18 mJ/m^2^ and the crosslinking process caused a decrease in this parameter, which is visible in ENR/DCP composites characterized by an SFE of 17.52 ± 1.08 mJ/m^2^. The presence of aromatic rings derived from quercetin in the ENR-50 matrix also significantly affected the SFE values. Due to the fact that the aromatic ring is closely associated with dispersive and hydrophobic forces, the presence of this functional group in quercetin structures caused a significant decrease in the SFE parameter of ENR composites, which amounted to a value of approximately 14.77–15.59 mJ/m^2^. Some of the polar -OH groups present in the quercetin structure participate in the crosslinking process and disappear; thus, the polar component becomes almost invisible.

The aging processes resulted in a change in SFE tested composite values with polar and dispersive components. The surface free energy rose dramatically after solar aging, particularly in the case of ENR-50-based composites containing silica. The stiff surface of ENR-50 composites with silica, similarly to ENR cured with DCP, had a highly crosslinked structure; thus, it was more vulnerable to solar radiation. On the other hand, the SFE of composites with only quercetin rose after thermo-oxidative aging. This happened because quercetin without silica, and other components, can only play one role in ENR-50 and acts as a crosslinker. Most -OH groups in quercetin are occupied and form crosslinks; thus are unable to fully protect the material from thermo-oxidation. The polar component increased in almost every case, particularly in ENR-50-based materials with quercetin, which also indicated the aging phenomenon. ENR-50 composites containing not only quercetin, but also silica, seemed to be most stable.

### 2.4. Color Identification

As mentioned above, quercetin can be recognized as a natural colorant [6], and in some cases, as an indicator of aging [14]. According to color results presented in Figure 5, it can be noted that composites containing quercetin are characterized by a lower whiteness index, a higher chroma and a lower hue angle. The addition of silica, similarly to dicumyl peroxide, did not change the presented parameters. On the other hand, quercetin was able not only to crosslink the ENR-50 matrix, but also color it, which is also visible in Appendix A.

Furthermore, the aging processes caused significant color fluctuations in composites with quercetin. The observer recognized different colors when the color change index (dE) was above 5. As can be noted, the highest color change index was detected in the ENR/Q4 composite, where dE equaled 11.1 ± 0.6 after the solar aging and 11.9 ± 0.6 after thermo-oxidation processes. The color of composites without quercetin was more stable and was below 5.

### 2.5. FT-IR Absorbance Spectra Analysis

The chemical changes of the composite’s surface, which occurred during the aging processes, were not favorable. All of them indicated that the tested materials were not stable or were resistant to different environmental conditions. The most common chemical modifications of the surface, which occurred during different aging processes, were identified between C-H moieties (~2800 cm^−1^) and C=O groups (~1700 cm^−1^) [20]. Observing the changes in the oxirane groups, which occurred in 875 cm^−1^, is also important in the case of ENR-based materials. According to the spectra presented in Figure 6 and Figure 7, the most significant changes were detected at a wavenumber of 3400 cm^−1^, which represents O-H bonding [21]. This signal appearing after aging processes is usually related to the formation of unstable hydroperoxides, carboxylic acids or alcohols, as a result of the oxidation process of the material [22], but is also present in quercetin structures, as well as silica. In ENR-based materials, such groups are able to occur after joining oxygen to the polymer chain or by opening oxirane rings. The least significant changes occurred in the case of ENR/quercetin4 and ENR/quercetin2/silica15 composites. All O-H groups located in these materials before the aging processes belong to the quercetin structure. The increased intensity of oxirane groups can also indicate the materials’ aging, which is visible in Figure 8. According to this figure, the most stable composites are ENR/Q4 and ENR/SIL/Q2, because the changes in oxirane rings are invisible. The structure of ENR-50 is presented in Figure 9 and the proposed mechanism of free radical scavenging activity of quercetin connected to ENR-50 chain is illustrated in Figure 10.

## 3. Materials and Methods

### 3.1. Materials and Processing

The investigated materials were prepared using epoxidized natural rubber with 50% epoxidation available commercially from the Muang Mai Guthrie Public Limited Company of Thailand (named Dynathai Epoxyprene 50 (ENR-50)). The crosslinking agent and anti-aging substance was quercetin hydrate, with ≥95% purity, produced by Sigma Aldrich (Munich, Germany). The added filler, a hydrophilic fumed silica (Aerosil^®^ 380), characterized by a specific surface area of 380 m^2^/g, obtained from Evonik Operations GmbH (Essen, Germany), was utilized to improve mechanical properties. The referential sample was crosslinked using a conventional curing agent–dicumyl peroxide (DCP, bis (α, α-dimethylbenzyl) peroxide) with 98% of purity purchased from Merck (Darmstadt, Germany).

First, the selected substances, according to the composition presented in Table 2, were mixed using a laboratory micromixer (Brabender Lab-Station from Plasti-Corder with the Julabo cooling system (Duisburg, Germany)) with a speed of 60 rpm for 15 min and at a temperature of about 25 °C to 70 °C, because of a spontaneous heating occurrence during the mixing process. Next, the obtained, uncured elastomers were processed at ambient temperature using a laboratory mixing mill characterized by friction of 1–1.2. The prepared materials were cured by an electrically heated hydraulic laboratory press (Skamet 54436, SKAMET, Skarzysko-Kamienna, Poland) for 20 min at T = 160 °C and *p* = 14 MPa using steel vulcanization molds placed between the shelves of the heated press. The polytetrafluoroethylene (PTFE, Teflon^®^) films provided by Holtex® (Rzgow, Poland) were used to avoid the phenomenon of adhesion between samples and molds. The dimensions of obtained samples were as follows: a length of 120 mm, a width of 80 mm and a thickness of approximately 1 mm. The pressed samples were subsequently tested, aged and retested to estimate the aging resistance.

### 3.2. Thermogravimetry Analysis

Thermogravimetric analysis (TGA) was applied to assess the thermal resistance of the tested materials. The measurement was conducted using a TGA/DSC1 STARe System equipped with a Gas Controller GC10^®^ device (Greifensee, Switzerland). Indium and zinc were used as standards to calibrate the equipment. The test was performed with a heating rate of 10 °C/min in a temperature range of 25–800 °C. In the first stage of heating (from 25 °C to 700 °C), argon, with a flow rate of 60 mL/min, was applied, and during the heating process, from 700 °C to 800 °C, air with the same flow rate was used. Polycrystal aluminum oxide crucibles with a volume of 70 μL were used in this measurement and the weight of the specimens amounted to approximately 7–9 mg.

### 3.3. Solar Aging Process

The solar aging process was conducted in an Atlas SC 340 MHG Solar Simulator climate chamber (AMETEK Inc., Berwyn, IL, USA) equipped with a 2500 W MHG lamp. A special rare-earth halogen lamp gives a unique range of solar radiation. The radiation intensity equaled 1200 W/m^2^ at 100% lamp power intensity. The samples were aged for 100 h at a low temperature of T = −10 °C, with maximal solar radiation during the whole aging process.

### 3.4. Thermo-Oxidative Aging Process

Thermo-oxidative aging was conducted based on PN-82/C-04216 Polish standard use laboratory Binder^®^ from an FD series dryer (Tuttlingen, Germany) at a temperature of T = 70 °C for 100 h. The dryer with forced convection exposed the provided samples to air at a raised temperature, enabling the oxidation process to occur.

### 3.5. Static Mechanical Tests

The pressed ENR-50-based materials were formed by cutting them into dumbbell-shaped samples (type 2) based on ISO 37 standards (samples’ thicknesses were 1 mm, their lengths were 75 mm and their widths were 12.5 mm). Static mechanical tests were carried out in accordance with ISO 37 standards before and after different aging processes utilizing the Zwick 1435 mechanical testing machine (Zwick Roell GmbH & Co. KG, Ulm, Germany). The crosshead speed during measurement amounted to 500 mm/min. Next, the aging coefficient was calculated from tensile strength results (TS) and elongation at break (E_b_) results obtained before and after different aging processes based on Equation (1).
(1)$$A_{f}\;\left[\%\right]=\frac{TS_{aftr\;aging}*Eb_{after\;aging}}{TS_{before\;aging}*Eb_{before\;aging\;}}$$
where:$A_{f}\left[\%\right]$—aging coefficient [MPa];${\mathrm{TS}}_{\mathrm{after}\;\mathrm{aging}}\left[\mathrm{MPa}\right]$—composite’s tensile strength after aging process (TS) [MPa];${\mathrm{Eb}}_{\mathrm{after}\;\mathrm{aging}}\left[\%\right]$—composite’s elongation at break after aging process (Eb) [%];${\mathrm{TS}}_{\mathrm{before}\;\mathrm{aging}}\left[\mathrm{MPa}\right]$—composite’s tensile strength before aging process (TS) [MPa];${\mathrm{Eb}}_{\mathrm{before}\;\mathrm{aging}}\left[\%\right]$—elongation at break before aging process (Eb) [%];

### 3.6. Contact Angle and Surface Free Energy Measuements

Contact angles (θ_C_) for the tested biocomposites were measured before and after the aging processes by means of an OCA 15EC goniometer from DataPhysics Instruments GmbH (Filderstadt, Germany) equipped with software module SCA 20, using three types of liquid characterized by different polarities: water, diiodomethane and ethylene glycol. For every composite, at least six CA results were obtained. The Braun DS-D 1000 SF syringe with a needle characterized by an outer diameter of OD = 0.52 mm and inner diameter of L = 0.25 mm and a length of 38.10 mm was applied. The volume of liquid drops amounted to 1 μL. Based on the data obtained, the surface free energy (SFE) [mJ/m^2^] was calculated using the Owens–Wendt–Rabel–Kaelble (OWRK) model, where the geometric mean of dispersive and polar components of the liquid’s surface tension ($\sigma_{L}^{d}$ and $\sigma_{L}^{p}$) and the same components of the solid’s surface energy ($\sigma_{s}^{p}$ and $\sigma_{s}^{d}$) are included and showed in Equation (2):(2)$$\sigma_{SL}=\sigma_{S}+\sigma_{L}-2\sqrt{\sigma_{s}^{d}\sigma_{L}^{d}}-2\sqrt{\sigma_{s}^{p}\sigma_{L}^{p}}$$
where:$\sigma_{\mathrm{SL}}$—interfacial tension at the border of solid and liquid $\left[\frac{\mathrm{mJ}}{m^{2}}\right]\mathrm{or}\left[\frac{\mathrm{mN}}{m}\right]$;$\sigma_{S}$—surface energy of solid $\left[\frac{\mathrm{mJ}}{m^{2}}\right]\mathrm{or}\;\left[\frac{\mathrm{mN}}{m}\right]$;$\sigma_{L}$—surface tension of liquid $\left[\frac{\mathrm{mJ}}{m^{2}}\right]\mathrm{or}\;\left[\frac{\mathrm{mN}}{m}\right]$;$\sigma_{s}^{d}$—dispersive component of solid’s surface tension $\left[\frac{\mathrm{mJ}}{m^{2}}\right]\mathrm{or}\;\left[\frac{\mathrm{mN}}{m}\right]$;$\sigma_{L}^{d}$—dispersive component of liquid’s surface tension $\left[\frac{\mathrm{mJ}}{m^{2}}\right]\mathrm{or}\;\left[\frac{\mathrm{mN}}{m}\right]$;$\sigma_{s}^{p}$—polar component of solid’s surface energy $\left[\frac{\mathrm{mJ}}{m^{2}}\right]\mathrm{or}\;\left[\frac{\mathrm{mN}}{m}\right]$;$\sigma_{L}^{p}$—polar component of liquid’s surface tension $\left[\frac{\mathrm{mJ}}{m^{2}}\right]\mathrm{or}\;\left[\frac{\mathrm{mN}}{m}\right]$;

The presented expression (Equation (2)) was substituted in the Young equation (Equation (3)), where the polar and the dispersive components of the solid’s surface energy were determined from the regression line in a suitable plot.
(3)$$\sigma_{L}cos\theta_{C}=\sigma_{S}-\sigma_{SL}$$

### 3.7. Color Identification

The color of the prepared composites was measured using a Spectrophotometer UV-VIS CM-36001 (Konica Minolta Sensing, Inc., Osaka, Japan) based on the PN-EN ISO 105-J01 standard. The signal reflected from the surface of the sample is detected and converted into a color impression perceivable by the human eye. The results were presented using the CIE-Lab system, where the L-axis describes lightness, the a-axis is responsible for red–green tone expression and the b-axis corresponds to yellow–blue colors. Based on the a, b and L results, the color difference (ΔE) between aged and unaged samples was calculated according to Equation (4). Furthermore, the whiteness index (W_i_), chroma (C_ab_) and hue angle (h_ab_) values of aged and unaged materials were calculated based on Equations (5)–(7).
(4)$$\Delta E=\sqrt{\Delta a^{2}+\Delta b^{2}+\Delta L^{2}}$$
(5)$$W_{i}=100-\sqrt{a^{2}+b^{2}+{\left(100-L\right)}^{2}}$$
(6)$$C_{\mathrm{ab}}=\sqrt{a^{2}+b^{2}}$$
(7)$$h_{\mathrm{ab}}\{\begin{array}{c} \mathrm{arctg}\left(\frac{b}{a}\right),\;\mathrm{when}\;a>0\;\cap b>0\\ 180{}^{\circ}+\mathrm{arctg}\left(\frac{b}{a}\right),\;\mathrm{when}\;\left(a<0\cap b>0\right)\cup (a<0\cap b<0)\\ 360{}^{\circ}+\mathrm{arctg}\left(\frac{b}{a}\right),\;\mathrm{when}\;a>0\cap b<0 \end{array}$$
where:Δa—difference of a parameter between aged and unaged samples;Δb—difference of b parameter between aged and unaged samples;ΔL—difference of L parameter between aged and unaged samples.

### 3.8. FT-IR Absorbance Spectra Analysis

The changes, which occurred on the surface of the tested biomaterials during the aging processes were analyzed by means of Thermo Scientific Nicolet 6700 Fourier transform infrared spectroscopy (FT-IR) device connected to diamond Smart Orbit ATR sampling equipment (Thermo Fischer Scientific Instruments, Waltham, MA, USA). The spectra were obtained in the absorption mode using 64 scans and resolution of 4 cm^−1^. The range of wave number amounted to 4000–400 cm^−1^.

## 4. Conclusions

Based on mechanical and FT-IR results, fabricated ENR-50-based composites crosslinked with silica, or a combination of quercetin and silica, are more resistant to aging processes than conventionally cured ENR-50 using dicumyl peroxide (DCP). Nevertheless, applying only quercetin or only silica as a crosslinking agent is not effective from the mechanical point of view as well as considering the material’s aging and thermal stability. Therefore, an additional combination of quercetin and silica in the proper proportions is important not only to improve resistance to different environmental conditions, but also to obtain better mechanical properties with enhanced thermal stability. The best compromise between aging resistance and tensile strength was obtained for the ENR/SIL/Q2 composite, where the aging coefficient amounted to approximately 1 and TS equaled above 3 MPa. Furthermore, FT-IR spectra revealed only irrelevant changes in this composite’s surface, so its chemical structure is the most stable. Such composites, due to their natural origin, seem to be environmentally friendly, safe for human beings, and have the potential to be dedicated to biomedical applications. Nevertheless, more advanced studies must be performed, particularly from a medical point of view.

## Figures and Tables

**Figure 1 ijms-22-10894-f001:**
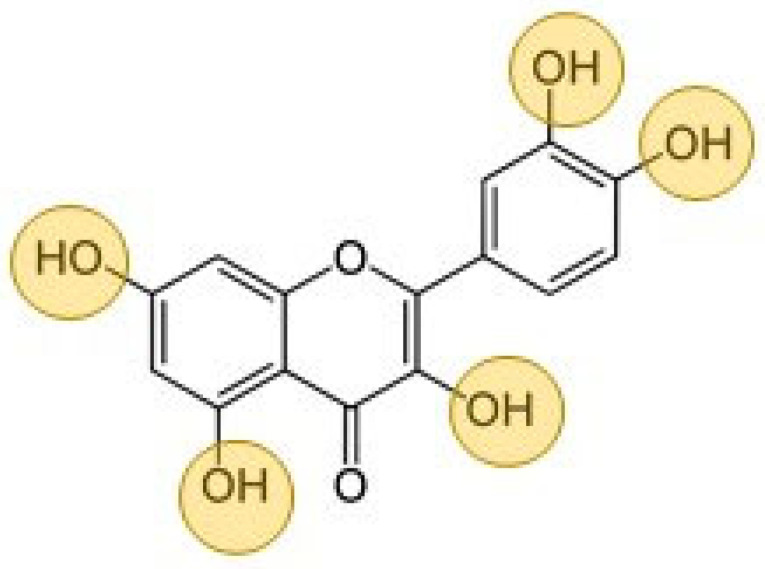
The structure of quercetin containing five hydroxyl groups responsible for antioxidant and crosslinking properties.

**Figure 2 ijms-22-10894-f002:**
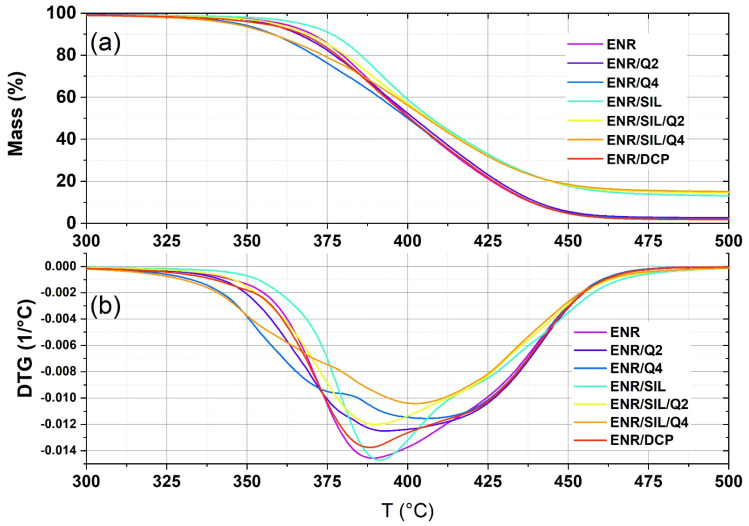
TGA (**a**) and DTG (**b**) curves of uncured ENR-50 and ENR-based composites crosslinked with quercetin, silica and combination of silica and quercetin.

**Figure 3 ijms-22-10894-f003:**
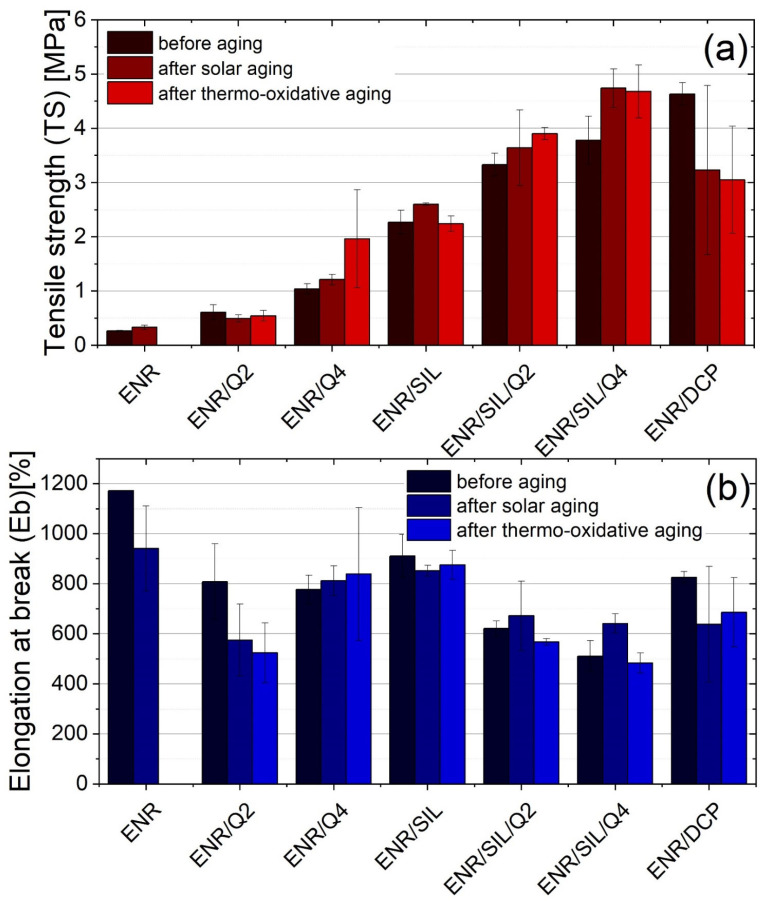
Tensile strength results (TS) (**a**), elongation at break (**b**).

**Figure 4 ijms-22-10894-f004:**
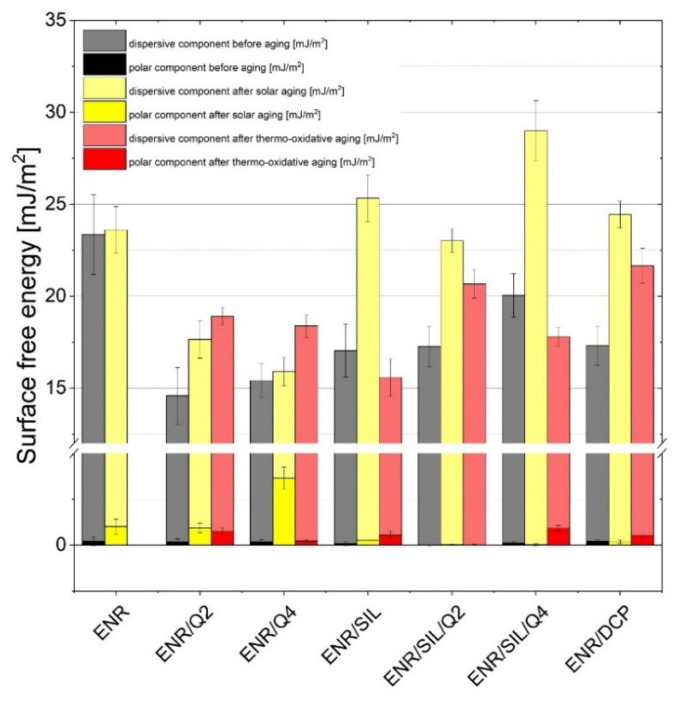
Surface energy results of ENR-50-based composites before and after aging processes.

**Figure 5 ijms-22-10894-f005:**
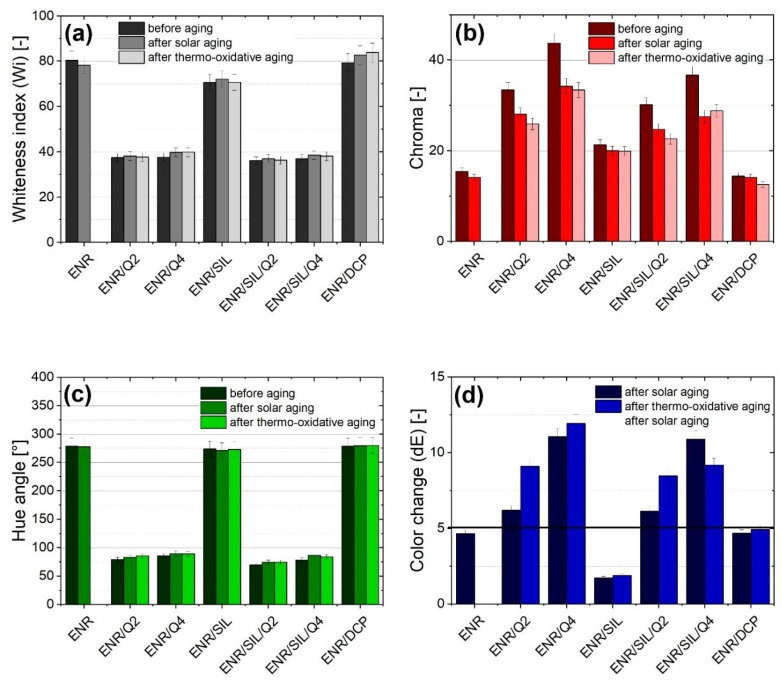
The whiteness index (**a**), chroma values (**b**), hue angles (**c**) and color change results (**d**) of ENR-50-based composites.

**Figure 6 ijms-22-10894-f006:**
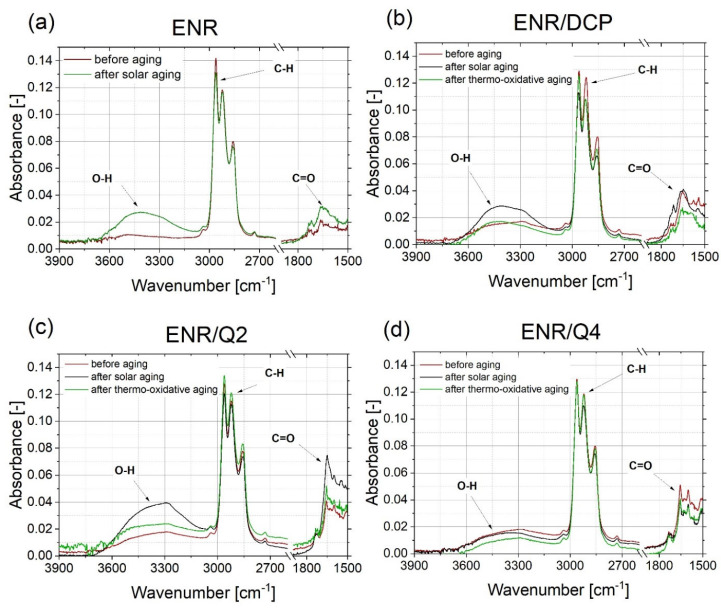
FT-IR spectra of uncured ENR (**a**), ENR cured with DCP (**b**), ENR cured with 2 phr of quercetin (**c**) and ENR cured with 4 phr of quercetin (**d**) before and after solar as well as thermo-oxidative aging.

**Figure 7 ijms-22-10894-f007:**
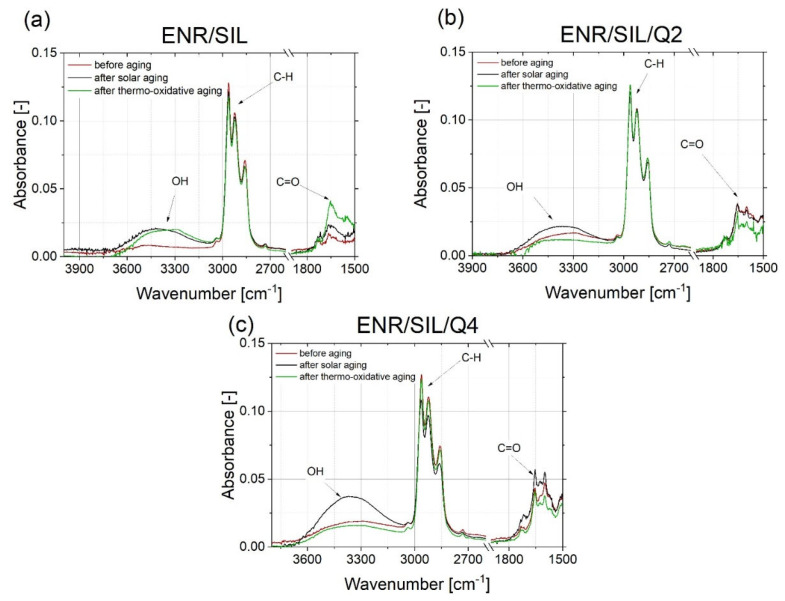
FT-IR spectra of ENR with silica (**a**), ENR cured with combination 2 phr of quercetin and silica (**b**), ENR cured with 4 phr of quercetin and silica (**c**) before and after solar as well as thermo-oxidative aging.

**Figure 8 ijms-22-10894-f008:**
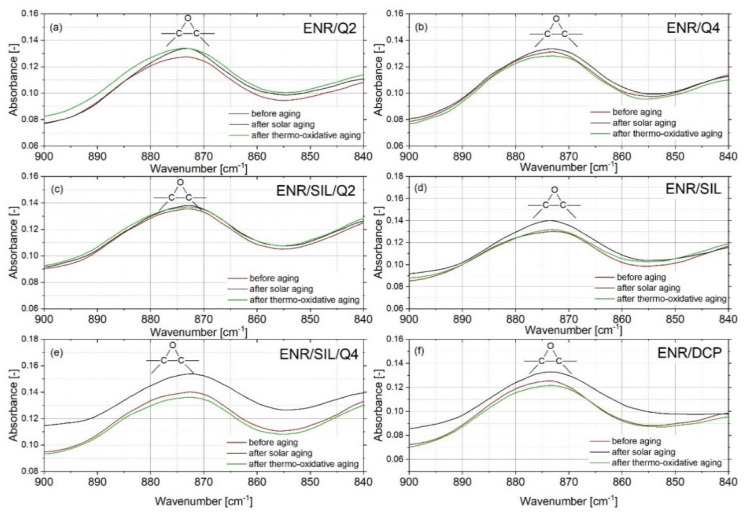
FT-IR spectra near oxirane rings of ENR with 2 phr of quercetin (**a**), ENR cured with 4 phr of quercetin (**b**), ENR cured with combination of 2 phr of quercetin and silica (**c**), with silica (**d**), with combination of 4 phr of quercetin and silica (**e**) and ENR-50 cured with DCP before and after solar as well as thermo-oxidative aging (**f**).

**Figure 9 ijms-22-10894-f009:**
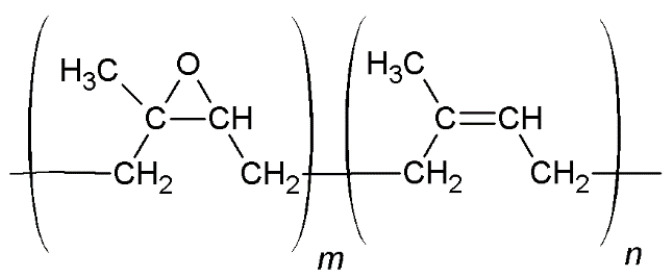
Chemical structure of epoxidized natural rubber (ENR-50).

**Figure 10 ijms-22-10894-f010:**
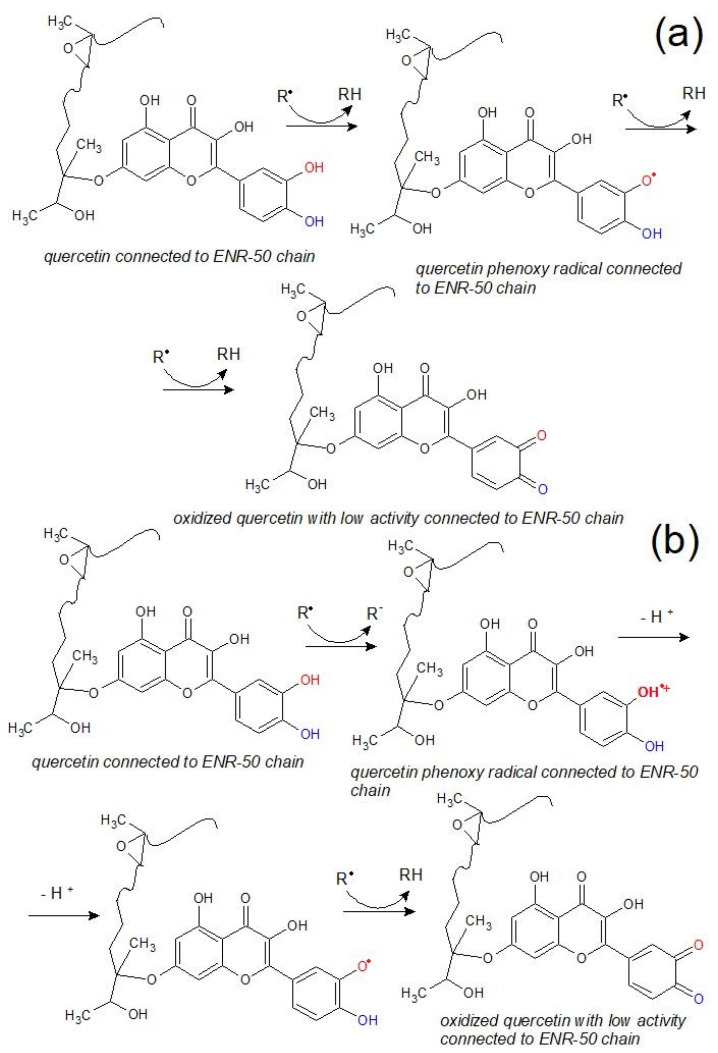
Proposed mechanisms for free radical scavenging activity of quercetin connected to ENR-50 chain. Hydrogen atom donation (**a**) and electron transfer-proton transfer (**b**).

**Table 1 ijms-22-10894-t001:** Temperatures at maximum mass loss rate of the ENR-50-based composites and the residual rate [%].

Sample	T_2%_ [°C]	Residual Rate [%] (T = 800 °C)
ENR	343	0.96
ENR/DCP	332	0.41
ENR/SIL	350	12.29
ENR/Q2	342	1.57
ENR/Q4	328	0.27
ENR/SIL/Q2	340	12.46
ENR/SIL/Q4	325	12.17

**Table 2 ijms-22-10894-t002:** Composition of tested ENR-50-based materials.

Components	Mass Ratio [phr*]						
	ENR	ENR/DCP	ENR/SIL	ENR/Q2	ENR/Q4	ENR/SIL/Q2	ENR/SIL/Q4
ENR-50	100	100	100	100	100	100	100
DCP	-	2	-	-	-	-	-
quercetin	-	-	-	2	4	2	4
silica	-	-	15	-	-	15	15

* phr—parts per hundred rubber.

## Data Availability

Data sharing is not applicable for this article.

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
