# Peer review of "Aging Resistance of Biocomposites Crosslinked with Silica and Quercetin"

_ijms, 2021, doi:10.3390/ijms221910894_

Round 1

Reviewer 1 Report

See attached file

Author Response

Institute of Polymer and Dye Technology

Technical University of Lodz

90-924 Lodz, ul Stefanowskiego 12/16, Poland

Tel.: +48 42 631 32 23, Fax: +48 42 636 25 43

September 17, 2021

Materials

Dear Professor,

We are resubmitting our revised paper entitled Aging resistance of biocomposites crosslinked with combination of silica and quercetin by Anna Masek and Olga Olejnik with a request to reconsider it for publication in International Journal of Molecular Sciences (IJMS).

We have carefully considered the Editor and Reviewers' comments. The manuscript was revised exactly according to these comments. The list of responses to the reviewers’ comments and corrections made in the manuscript is attached.

The manuscript has not been previously published, is not currently submitted for review to any other journal, and will not be submitted elsewhere before a decision is made by this journal.

For correspondence please use the following information:

corresponding author: Anna Masek

Institute of Polymer and Dye Technology

Technical University of Lodz

90-924 Lodz, ul Stefanowskiego 12/16, Poland

Tel.: +48 42 631 32 93

Fax: +48 42 636 25 43

e-mail: anna.masek@p.lodz.pl

Yours sincerely,

Ph. D., D.Sc. Anna Masek

Reviewer #1:

The entire manuscript needs to be reviewed for English grammar, as it cannot be published in its present state.

Answer 1 for Reviewer #1:

We thank the Reviewer for paying attention to English grammar. We have improved our manuscript and corrected grammatical errors.

Reviewer #1:

No previous proof or investigation into potential 'self-healable' properties in these materials are presented. As it is a central part of the title, this is highly misleading.

Answer 2 for Reviewer #1:

We are thankful for the Reviewer’s comment. We proved and presented potential 'self-healable' properties in these materials in our previous work entitled “Self-healable biocomposites crosslinked with combination of silica and quercetin”. Nevertheless, we agree that the title of current paper might be misleading, therefore we decided to change the title from “Aging resistance of self-healable biocomposites crosslinked with combination of silica and quercetin” into “Aging resistance of biocomposites crosslinked with combination of silica and quercetin”

Reviewer #1:

Missing ENR thermos-oxidative aging control sample? (see Figure 3, and subsequent Figures). Why? This is a key piece of information in interpreting the subsequent results.

Answer 3 for Reviewer #1:We appreciate the Reviewer’s suggestions. Nevertheless, we mentioned in our paper that the ENR thermo-oxidative aging control sample could not be tested because of its degradation. The material was too sticky and further investigation was impossible.  However, we noticed that ENR/DCP sample results were missing in Figure 3 and we corrected this mistake.

Figure 3. Tensile strength results (TS) (a), elongation at break (b).

Reviewer 2 Report

The entire manuscript needs to be reviewed for English grammar, as it cannot be published in its present state.

No previous proof or investigation into potential 'self-healable' properties in these materials are presented. As it is a central part of the title, this is highly misleading.

Missing ENR thermos-oxidative aging control sample? (see Figure 3, and subsequent Figures). Why? This is a key piece of information in interpreting the subsequent results.

Author Response

Institute of Polymer and Dye Technology

Technical University of Lodz

90-924 Lodz, ul Stefanowskiego 12/16, Poland

Tel.: +48 42 631 32 23, Fax: +48 42 636 25 43

September 17, 2021

Materials

Dear Professor,

We are resubmitting our revised paper entitled Aging resistance of biocomposites crosslinked with combination of silica and quercetin by Anna Masek and Olga Olejnik with a request to reconsider it for publication in International Journal of Molecular Sciences (IJMS).

We have carefully considered the Editor and Reviewers' comments. The manuscript was revised exactly according to these comments. The list of responses to the reviewers’ comments and corrections made in the manuscript is attached.

The manuscript has not been previously published, is not currently submitted for review to any other journal, and will not be submitted elsewhere before a decision is made by this journal.

For correspondence please use the following information:

corresponding author: Anna Masek

Institute of Polymer and Dye Technology

Technical University of Lodz

90-924 Lodz, ul Stefanowskiego 12/16, Poland

Tel.: +48 42 631 32 93

Fax: +48 42 636 25 43

e-mail: anna.masek@p.lodz.pl

Yours sincerely,

Ph. D., D.Sc. Anna Masek

Reviewer #2:

Comments and Suggestions for Authors

This manuscript is in continuation of an earlier work Ref#18, showing the antioxidant behavior of added silica and quercetin. I have some major comments:

  1. There is no need to highlight self-healing since in this manuscript it is not assessed.

Answer 1 for Reviewer #2:

We are thankful for the Reviewer’s comment. We proved and presented potential 'self-healable' properties in these materials in our previous work entitled “Self-healable biocomposites crosslinked with combination of silica and quercetin”. Nevertheless, we agree that the title of current paper might be misleading, therefore we decided to change the title from “Aging resistance of self-healable biocomposites crosslinked with combination of silica and quercetin” into “Aging resistance of biocomposites crosslinked with combination of silica and quercetin”

Reviewer #2:

  1. Please show the structure of ENR and clarify the mechanism there.

Answer 2 for Reviewer #2:

We are grateful for the Reviewer’s suggestions, therefore we showed the ENR structure and we also presented the bonding between ENR-50 and quercetin as well as we showed the mechanism of quercetin, which prevents the oxidation process.

Figure 1. Chemical structure of epoxidized natural rubber (ENR-50).

Figure 2. Proposed mechanisms for free radical scavenging activity of quercetin connected to ENR-50 chain. Hydrogen atom donation (a) and electron transfer – proton transfer (b).

Reviewer #2:

  1. The Results and discussion part is too long. Please make it more concise and sharper to the point.

Answer 3 for Reviewer #2:

We appreciate Reviewer’s suggestions. We wanted to thoroughly explained and presented the obtained results, therefore this part paragraph to be long. Nevertheless, every part of “Results and discussion” paragraph has a summary in the end, which are concise and indicates the most relevant conclusions.

Reviewer #2:

  1. The TG apart for thermal stability is misleading. First, what is the application of this material? Are you going to use that at a temperature region above 300C? The mechanism for degradation at this temperature in an inert atmosphere is pyrolysis, how does the crosslinking density, etc affecting this mechanism?

Answer 4 for Reviewer #2:

We are grateful for Reviewer’s opinion. We changed the part concerning TG analysis and we focused on thermal stability. We wanted to show, that not only a small dose of quercetin is important as antioxidant and crosslinker but silica is needed to improve the mechanical properties and thermal stability. The crosslinking density of composite after TG is not possible to identify, because of the small amount of this material. Such material can be dedicated to biomedical application and using it above 300C is not necessary but also not  out of the question.

Reviewer #2:

  1. For antioxidant behavior, the OIT (oxidation induction time) is the most relevant method to assess not the mechanical properties. The authors need to show that one!

Answer 5 for Reviewer #2:

We thank the Reviewer for this suggestion. However, time for responding all the reviews is too short to prepare all the required measurements because of the limit in device availability. We showed the behavior of tested composites before and after aging process, including thermo-oxidative aging, where the samples were exposed to oxygen. Not only mechanical properties were showed, but also other results, including FT-IR spectra before and after different aging processes, were analyzed.

  1. Increased tensile strength could be related to crosslinking due to aging. This mechanism should be highlighted here.

Answer 6 for Reviewer #2:

We appreciate Reviewer’s comment. Nevertheless, we presented structure of crosslinked ENR-50 by using combination of quercetin and silica in our previous paper and we only mentioned the crosslinking process due to aging in the current article.

Figure 3. Proposed structure and interactions in ENR/quercetin/silica composite.

Reviewer 3 Report

This manuscript is in continuation of an earlier work Ref#18, showing the antioxidant behavior of added silica and quercetin. I have some major comments:

  1. There is no need to highlight self-healing since in this manuscript it is not assessed.
  2. Please show the structure of ENR and clarify the mechanism there.
  3. The Results and discussion part is too long. Please make it more concise and sharper to the point.
  4. The TG apart for thermal stability is misleading. First, what is the application of this material? Are you going to use that at a temperature region above 300C? The mechanism for degradation at this temperature in an inert atmosphere is pyrolysis, how does the crosslinking density, etc affecting this mechanism?
  5. For antioxidant behavior, the OIT (oxidation induction time) is the most relevant method to assess not the mechanical properties. The authors need to show that one!
  6. Increased tensile strength could be related to crosslinking due to aging. This mechanism should be highlighted here.     

Author Response

Institute of Polymer and Dye Technology

Technical University of Lodz

90-924 Lodz, ul Stefanowskiego 12/16, Poland

Tel.: +48 42 631 32 23, Fax: +48 42 636 25 43

September 17, 2021

Materials

Dear Professor,

We are resubmitting our revised paper entitled Aging resistance of biocomposites crosslinked with combination of silica and quercetin by Anna Masek and Olga Olejnik with a request to reconsider it for publication in International Journal of Molecular Sciences (IJMS).

We have carefully considered the Editor and Reviewers' comments. The manuscript was revised exactly according to these comments. The list of responses to the reviewers’ comments and corrections made in the manuscript is attached.

The manuscript has not been previously published, is not currently submitted for review to any other journal, and will not be submitted elsewhere before a decision is made by this journal.

For correspondence please use the following information:

corresponding author: Anna Masek

Institute of Polymer and Dye Technology

Technical University of Lodz

90-924 Lodz, ul Stefanowskiego 12/16, Poland

Tel.: +48 42 631 32 93

Fax: +48 42 636 25 43

e-mail: anna.masek@p.lodz.pl

Yours sincerely,

Ph. D., D.Sc. Anna Masek

Reviewer #3:

The paper deals with the development of biocomposites based on epoxidized natural rubber

crosslinked with combination of silica and quercetin.

Due to the properties of the two components, it is expected that silica contributes to the

enhancement of both thermal stability and some mechanical properties (as evidenced in Figures 2 and 3) whereas quercetin, because of its antioxidant activity, let the aging resistance to be improved.

This is not clear from the text in “results and discussion”. The whole section is quite confusing and, in some parts contradictory (especially the sub-section 2.1 - hermogravimetry analysis). It should be re-written in a more clear and understandable way.

Answer 1 for Reviewer #3:

We thank Reviewer for the comment. We decided to re-write sub-section 2.1. (thermogravimetry) and refer to some specific comments below:

Reviewer #3:

Some specific comments:

Section 2.1 - Thermogravimetry analysis

  1. 3 lines 96-100 – The authors affirm than ENR/SIL/Q2 shows better thermal stability than ENR/SIL/Q4. Two paragraphs later (p. 3 lines 103-105) they state that ENR/Q4 and ENR/SIL/Q4 are characterized by the highest temperature at maximum mass loss rate, suggesting an increase of thermal stability of these two samples with respect to the others. This appears quite contradictory. The authors must define what they mean by “thermal stability” and, considering this, re-discuss the results.

Answer 2 for Reviewer #3:

We are thankful for Reviewer’s suggestions.  We define “thermal stability” as the start of the material’s decomposition in higher temperatures. We decided to remove the information about “the highest temperature at maximum mass loss rate” to avoid confusion.

Reviewer #3:

Figure 2 - The authors affirm that quercetin improves the thermal stability of ENR samples, but figure 2 shows that the thermal stability of ENR-based samples is enhanced, in terms of mass loss, by silica and not by quercetin (as also stated in the conclusions).

Answer 3 for Reviewer #3:

We appreciate the Reviewer’s comments and we decided to underline the information that silica is responsible for thermal stability. As mentioned above, we decided to remove the misleading information about “the highest temperature at maximum mass loss rate”, which did not indicate thermal stability.

Reviewer #3:

Table 3 – Please specify the temperature or temperature range at which the Residual rate (%) is measured.

Answer 4 for Reviewer #3:

We thank the Reviewer for the suggestion and we completed the missing information about the temperature at which the residual rate [%] is measured. The residual rate [%] was calculated in the end of the measurement at the temperature of 800°C.

Reviewer #3:

Materials and Methods

Section 4.1 – Thermogravimetry analysis

The TGA used to perform the experiments is from Mettler Toledo or from TA Instruments? The two Companies are competitors. Due to the location (Greifensee, Switzerland) I guess is from Mettler Toledo. Please correct at p. 10, line 270.

Answer 5 for Reviewer #3:

We are grateful for paying attention to this mistake. We have rectified this error: “The measurement was conducted using TGA/DSC1 STARe System equipped with a Gas Controller GC10 ® device (Greifensee, Switzerand)”

Reviewer #3:

English should be revised.

Answer 6 for Reviewer #3:

We thank the Reviewer for paying attention to English grammar. We have improved our manuscript and corrected grammatical errors.

Round 2

Reviewer 1 Report

The authors answered to all my questions and revised the text taking into account my suggestions and comments. Thus, the paper can be published in the present form

Author Response

Thank Yoy

Reviewer 2 Report

This looks much better now. I can accept this for publication as it stands. As a potential future improvement to the authors, consider trying to incorporate your (very good) molecular illustrations in a vector based format instead of a pixel based format.  

Author Response

Thank You

Reviewer 3 Report

The authors addressed almost all concerns from my side, however the OIT was not done. At least it is suggested to have that experiment as an outlook.

Author Response

Institute of Polymer and Dye Technology

Technical University of Lodz

90-924 Lodz, ul Stefanowskiego 12/16, Poland

Tel.: +48 42 631 32 23, Fax: +48 42 636 25 43

October 2, 2021

International Journal of Molecular Sciences (IJMS)

Dear Editor,

We are resubmitting our revised paper entitled Aging resistance of biocomposites crosslinked with combination of silica and quercetin by Anna Masek and Olga Olejnik with a request to reconsider it for publication in International Journal of Molecular Sciences (IJMS).

We have carefully considered the Editor and Reviewers' comments. The manuscript was revised exactly according to these comments. The list of responses to the editors’ comments and corrections made in the manuscript is attached.

The manuscript has not been previously published, is not currently submitted for review to any other journal, and will not be submitted elsewhere before a decision is made by this journal.

For correspondence please use the following information:

corresponding author: Anna Masek

Institute of Polymer and Dye Technology

Technical University of Lodz

90-924 Lodz, ul Stefanowskiego 12/16, Poland

Tel.: +48 42 631 32 93

Fax: +48 42 636 25 43

e-mail: anna.masek@p.lodz.pl

Yours sincerely,

Ph. D., D.Sc. Anna Masek

Editor:

  1. Authors need to provide SEM images or sample photos to show the morphology and/or microstructure of the biocomposites;

Answer 1 for Editor:

We thank Editor for paying attention to this issue. We provided SEM images of ENR-based samples in our previous article and we do not want to duplicate them. We presented microscopic images in supplementary material:

Figure S2. Microscope images of ENR-based composites: (a) pure uncurred ENR, (b) ENR cured with 15 phr of silica, (c) ENR cured with dicumyl peroxide (DCP), (d) ENR cured with 2 phr of quercetin, (e) ENR cured with 4 phr of quercetin, (f) ENR cured with 2 phr of quercetin and 15 phr of silica, (g) ENR cured with 4 phr of quercetin and 15 phr of silica.

Editor:

  1. Figure 3: Authors need to add representative stress-strain curves for the tensile tests;

Answer 2 for Editor:

We are thankful for this suggestion. We added representative stress-strain curves for the tensile stress in a supplementary material:

Figure S1. The strain-stress curve of tested ENR-based composites.

Editor:

  1. Figure 5: Authors need to add representative UV-VIS spectra to show how to obtain the

color results;

Answer 3 for Editor:

We are grateful for this suggestion. We added representative representative UV-VIS spectra in a supplementary material:

Figure S3. UV-VIS spectra of ENR-based composites before (green curve) and after solar aging (blue curve): (a) pure uncurred ENR, (b) ENR cured with 15 phr of silica, (c) ENR cured with dicumyl peroxide (DCP), (d) ENR cured with 2 phr of quercetin, (e) ENR cured with 4 phr of quercetin, (f) ENR cured with 2 phr of quercetin and 15 phr of silica, (g) ENR cured with 4 phr of quercetin and 15 phr of silica.

Editor:

  1. The resolution of all figures needs to be improved. 

Answer 3 for Editor:

We thank editor for this comment. We put our figures of improved resolution, nevertheless we need more precise criteria, for example how many dpi of resolution is needed.

Editor:

The authors addressed almost all concerns from my side, however the OIT was not done. At least it is suggested to have that experiment as an outlook.

Answer 4 for Editor:

The OIT measurement can not be realized now because of technical issues, however we want to perform this analysis in the future and the result will be presented in the next article.
